# Improving Irritable Bowel Syndrome (IBS) Symptoms and Quality of Life with Quebracho and Chestnut Tannin-Based Supplementation: A Single-Centre, Randomised, Double-Blind, Placebo-Controlled Clinical Trial

**DOI:** 10.3390/nu17030552

**Published:** 2025-01-31

**Authors:** Silvia Molino, Lorenza Francesca De Lellis, Maria Vittoria Morone, Marcello Cordara, Danaè S. Larsen, Roberto Piccinocchi, Gaetano Piccinocchi, Alessandra Baldi, Alessandro Di Minno, Hesham R. El-Seedi, Roberto Sacchi, Maria Daglia

**Affiliations:** 1R&D Unit, Silvateam S.p.A., 12080 San Michele Mondovì, Italy; 2Department of Pharmacy, University of Naples “Federico II”, Via D. Montesano 49, 80131 Naples, Italy; lo.delellis2@libero.it (L.F.D.L.); alessandra.baldi.alimenti@gmail.com (A.B.); alessandro.diminno@unina.it (A.D.M.); 3Department of Experimental Medicine, Section of Microbiology and Clinical Microbiology, University of Campania “L. Vanvitelli”, 80138 Naples, Italy; mariavittoria.morone@unicampania.it; 4School of Medicine, University of Milano-Bicocca, 20126 Milan, Italy; m.cordara@campus.unimib.it; 5School of Chemical Sciences, The University of Auckland, Auckland 1010, New Zealand; d.larsen@auckland.ac.nz; 6Level 1 Medical Director Anaesthesia and Resuscitation A. U. O. Luigi Vanvitelli, Via Santa Maria di Costantinopoli, 80138 Naples, Italy; roberto.piccinocchi@policliniconapoli.it; 7Comegen S.c.S., Società Cooperativa Sociale di Medici di Medicina Generale, Viale Maria Bakunin 41, 80125 Naples, Italy; gpiccino@tin.it; 8CEINGE-Biotecnologie Avanzate, Via Gaetano Salvatore 486, 80145 Naples, Italy; 9Department of Chemistry, Faculty of Science, Islamic University of Madinah, Madinah 42351, Saudi Arabia; elseedi_99@yahoo.com; 10Applied Statistic Unit, Department of Earth and Environmental Sciences, University of Pavia, Viale Taramelli 24, 27100 Pavia, Italy; roberto.sacchi@unipv.it; 11International Research Center for Food Nutrition and Safety, Jiangsu University, Zhenjiang 212013, China

**Keywords:** irritable bowel syndrome, tannins, quebracho, chestnut, food supplement, IBSS score, questionnaire for self-assessment of quality of life, randomised clinical trial

## Abstract

**Background/Objectives**: Irritable Bowel Syndrome (IBS) is a disorder of the gut-brain axis for which the gastroenterologist is most often consulted. Gastrointestinal symptoms and decreased quality of life lead to a considerable burden of disease. The exact causes of IBS are not well understood, and no standard therapy has been established. The primary outcome of the study focused on the improvements of the IBS symptoms assessed through the validated questionnaire IBS-Severity Scoring System (IBS-SSS). Similarly, secondary outcomes geared towards the improvement of the quality of life (IBS-Quality of Life (IBS-QoL) and the Gastrointestinal Quality of Life Index (GIQLI)) and specific IBS symptoms (bloating, abdominal distension, feeling of heaviness, abdominal pain, and flatulence), were assessed through self-administered questionnaires. Intestinal habits (consistency and frequency of depositions) through subject stratification into diarrhoea (IBS-D), constipation (IBS-C), mixed type (IBS-M), as well as the treatment tolerability were also evaluated. **Methods**: A randomised, placebo-controlled, double-blind, clinical trial was conducted on 156 enrolled IBS patients (79 female and 77 male), aged 18–70 years, randomised (1:1 allocation ratio) to receive either two capsules per day of the food supplement (containing 480 mg of a complex of tannin extracts) or a placebo for 56 days. Linear random intercept mixed models (LMM) were used to analyse all experimental variables **Results**: Supplementation resulted in a significant improvement (*p* < 0.05) in the primary outcome IBSS score, with respect to the placebo group, changing the IBS condition going from mild (242.3 ± 89.8) to moderate (148.1 ± 60.6). Similarly, all indicators concerning quality of life, and specific intestinal symptoms resulted in a significant improvement (*p* < 0.05). Furthermore, the tannin-based treatment showed the ability to modulate the response to different symptomatology such as diarrhoea and constipation, without side effects being reported. **Conclusions**: The use of a supplement based on chestnut and quebracho tannins presents great application potential in the management of IBS-related disorders, with the peculiarity of resolving opposite symptoms, such as diarrhoea and constipation, indiscriminately.

## 1. Introduction

One of the main reasons for consulting a gastroenterologist is irritable bowel syndrome (IBS), a chronic disorder of the gut–brain axis, often debilitating, characterised by alterations in bowel habits and recurrent abdominal pain, with a serious impact on the health-related quality of life [1].

This disorder is clinically diagnosed through the Rome IV criteria, developed by the international non-profit expert group “Rome Foundation”, which represents a basic tool commonly used by physicians to diagnose various functional gastrointestinal diseases on the basis of symptoms and clinical presentation. The diagnosis of IBS, according to this system, requires that a patient experience recurrent abdominal pain at least one day per week (on average) over the preceding three months, accompanied by at least two of the following: abdominal pain related to defecation, changes in stool frequency, and changes in stool form [2].

Four main subtypes of IBS are recognised, depending on the predominant stool form: diarrhoea-predominant IBS (IBS-D), constipation-predominant IBS (IBS-C), IBS with mixed bowel habits (IBS-M), and unclassified IBS (IBS-U) [1,2]. In general, female individuals and adults younger than 65 years are the main sufferers; the global prevalence has been estimated to be 11%, although it can vary greatly between countries depending on different lifestyles, diets, and diagnostic methods [3,4].

Although the aetiology of IBS is broad and not clearly understood, the most likely hypothesis is that the combination of multiple factors contributes to the development of this disorder, including microbiota dysbiosis, small intestinal bacterial overgrowth (SIBO), dysmotility, visceral hypersensitivity, mental disorders, altered communication between the gut and the brain, and psychosocial distress [5].

Gaps in the comprehensive understanding of the aetiology of IBS are reflected in a lack of clear and conclusive guidelines for its treatment. Several recent reviews, monographs, and position statements have tried to summarise the treatment recommendations, also differentiating for the different subtypes of IBS, including drugs, behavioural therapies, dietary supplements (i.e., probiotics and fibres), and targeted nutrition (i.e., low FODMAP diet poor in fermentable oligosaccharides, disaccharides, monosaccharides, and polyols) [1,4,5,6,7,8].

Given this framework, we considered the use of a formulation based on tannin phytocomplexes, since these secondary plant metabolites are known and extensively documented for their potential for managing multiple IBS-related impairments [9,10]. Alongside their known astringent properties, certain tannin-based plant extracts have shown a regulatory activity on the function and motility of the intestine, with effects on digestion, nutrient absorption and the formation of stools of better consistency [11,12,13]. Moreover, some authors reported the prebiotic effect through the modulation of the gut microbiota composition and functionality (i.e., synthesis of beneficial metabolites such as short-chain fatty acids (SCFAs)) [14,15,16].

The present study is a single-centre, randomised, double-blind, placebo-controlled clinical trial demonstrating the efficacy of a commercial product containing a tannin-based phytocomplex (Welltan^®^ Complex, Silvateam S.p.A, San Michele Mondoví, Italy) in improving the perceived gastrointestinal symptoms of IBS patients. Furthermore, a secondary goal was the evaluation of the quality of life as well as the product’s suitability in restoring the balance of intestinal function in the different IBS subtypes: IBS-D, IBS-C, IBS-M, and tolerability.

## 2. Materials and Methods

### 2.1. Food Supplement

The dietary supplement tested in the present clinical study is based on proprietary tannin phytocomplexes. More in detail, the quebracho colorado (Schinopsis lorentzii Engl.) wood extract is distinguished by its high condensed tannins content, especially profisetinidins and fisetin, among others [17]. The chestnut extract (*Castanea sativa* Mill.) is a phytocomplex characterised by ellagitannins, mainly of digalloyl-glucose, glucose, and gallic acid units [18]. The extraction methods were all food grade, characterised by a natural hot water extraction.

The food supplement containing Welltan^®^ Complex had previously been notified to the Italian Ministry of Health (notification number: 0019814).

Both the verum and the placebo, produced in compliance with European specifications on contaminants and microbiological limits, were provided free of charge by the INBB-Interuniversity Consortium of the National Institute of Biostructures and Biosystems, as sponsors of the clinical trial.

Subject supplementation consisted of two capsules per day of 240 mg containing each the functional raw material Welltan^®^ Complex. The placebo was provided under identical conditions and appearance, making the colour and taste indistinguishable from the food supplement.

### 2.2. Study Design

The effects of a tannin-based food supplement were assessed on IBS symptoms in adults with a moderate form of IBS according to Rome IV criteria [19] in a monocentric, randomised, placebo-controlled, parallel-group, double-blind clinical trial conducted by COMEGEN—Società Cooperativa Sociale (Naples, Italy). The enrolled participants were randomised into two experimental groups (treated group and placebo group with n = 78 per group). The dose forms were identical in terms of shape, colour, weight, and flavour, and the packaging was designed to make it impossible to distinguish between the food supplement and the placebo. Treatment allocation was not revealed to participants, clinicians, and investigators until the completion of the study to maintain the double blinding.

Before giving their written consent, participants were informed about this study both orally and in writing. The study protocol, the participants’ letter of intent, and the synopsis were approved by the Ethics Committee of Campania 1 (protocol number 5/23, 30 January 2024), and this study was conducted in accordance with the Helsinki Declaration of 1964 (as revised in 2000). Moreover, the clinical trial is listed on the ISRCTN registry under the ID number ISRCTN1681307 [20].

Enrolled subjects were randomly and unpredictably assigned to each of the two groups (n = 78, each) by simple randomisation (1:1 allocation ratio), treated either with food supplement (two capsules/day at lunch and dinner for a total dose of 480 mg/day) or the placebo. The randomisation was further stratified by three forms of IBS (constipated-IBS-C, diarrhoeal-IBS-D, or mixed-IBS-M). The clinical status of patients was evaluated at baseline (T0) and 56 days after the start of treatment through self-administered questionnaires: IBS-Severity Scoring System (IBS-SSS), IBS-Quality of Life (IBS-QoL), Gastrointestinal Quality of Life Index (GIQLI), Bristol Stool Form Scale (BSFS), and bowel movements/week (BM/wk) and IBS symptoms.

This study took five months to complete, including two months for patient enrolment and three months for the study, considering the timing for the last enrolled patient.

Patients did not follow any specific diet to evaluate only the effect induced by supplementation and were not allowed to take any dietary supplements. Any intake of drugs not listed in the exclusion criteria had to be communicated in advance and discussed with the clinician.

### 2.3. Participants and Recruiting

In total, 156 patients with IBS according to Rome IV criteria [19] were enrolled and then allocated into two groups. Subjects of both sexes and aged between 18 and 70 years were considered for this study, on condition that they had been presenting symptoms of IBS for at least three months (with onset at least six months earlier). The symptom pattern was required to be recurrent abdominal pain on at least one day a week, associated with two or more of the following criteria: symptoms associated with the act of defecation and/or associated with a change in the frequency of bowel movements and/or associated with a change in the shape of stools, as described by the Rome IV diagnostic criteria [19]. The forms of IBS considered were the predominantly constipated form (with the following stool consistency: more than 25% hard stools and less than 25% soft stools), the predominantly diarrhoeal form (with the following stool consistency: more than 25% soft stools and less than 25% hard stools) and the mixed form. The three forms were stratified into the two experimental groups to obtain a balance, although the IBS-D form was the form with the highest incidence in the population.

All subjects with intestinal symptomatology that did not meet the criteria described above were not considered. None of the enrolled subjects were aged <18 and >70; none were pregnant or lactating; and none were taking substances of abuse (i.e., caffeine, smoke, alcohol, drugs) or were on medication including opioid drugs or other drugs that have a significant impact on intestinal function. Subjects who had taken antibiotics within the previous four weeks or in the last six months (based on the intensity and duration of antibiotic treatment) were excluded.

Patients meeting one of the following exclusion criteria were also not considered eligible: previous gastrointestinal tract surgery, Parkinson’s and Alzheimer’s disease, HIV-acquired immunodeficiency, non-self-sufficiency, presence of cognitive disorders that may hinder the response to questionnaires, lack of a propensity to collaborate, difficulty in going to the reference facility on time, allergy to the ingredients of the experimental products (active-placebo). Moreover, if subjects were diagnosed with other pathologies, they were not considered incompatible by the investigating doctor.

### 2.4. Primary Outcome

The primary outcome was to evaluate the efficacy of the food supplement in the general improvement of perceived gastrointestinal symptoms. Regarding enteric symptomatology, the assessment method was the validated questionnaire for the self-assessment of the severity of gastrointestinal symptoms characterising IBS, IBS-Severity Scoring System (IBS-SSS). The questionnaire consists of 5 items referring to abdominal pain, bowel dysfunction, and general well-being. The sum of the partial scores allows discriminating the severity of IBS-related symptoms into three levels: mild (from 75 to <175), moderate (from 175 to <300), or severe (>300).

### 2.5. Secondary Outcomes

The secondary outcomes included the analysis of any improvement in the quality of life of the subjects in response to the experimental treatment, which was carried out through two questionnaires: a questionnaire for self-assessment of quality of life in the individual with IBS (i.e., IBS-Quality of Life (IBS-QoL)) and a questionnaire for self-assessment of quality of life associated with gastrointestinal symptoms (i.e., Gastrointestinal Quality of Life Index (GIQLI)). Both questionnaires consider, through different sets of questions, different aspects of the subject’s life by assessing from physical symptomatology to interpersonal relationships and the emotional sphere. The self-assessment IBS-QoL consider 8 items: dysphoria, interference with daily activities, body image, health concerns, abstention from food, social sphere, sexual sphere, and interpersonal relationships. The GIQLI questionnaire evaluates symptoms, emotions, physical condition, social integration, and the effect of medical treatment. In both cases, lower scores correspond to lower quality of life, while higher scores indicate higher standards of life.

Moreover, the bowel function diary filled by patients was used to evaluate the improvement of stool consistency and bowel movements, together with IBS characteristic symptoms. The first outcome was assessed with the Bristol Stool Form Scale (BSFS), which rates stool consistency on a spectrum of seven types, from harder and lumpy (lower scores) to soft and watery (higher scores). The number of weekly bowel movements (bowel movements/week (BM/wk)) was reported by the subject in the bowel function diary. Moreover, the symptoms of bloating, abdominal distension, feeling of heaviness, abdominal pain and flatulence were rated by filling in a 5-point Likert scale (0 for no discomfort and 5 for maximum discomfort).

### 2.6. Tolerability

The participants were closely observed for the occurrence of any unfavourable side effects from the treatment. Individuals who were allergic to any of the ingredients in the food supplement were excluded from the trial.

### 2.7. Statistical Analysis

The sample size calculation involved recruiting 148 subjects in total, to ensure a to ensure 95% power and alpha significance of 0.05, with a small-to-medium effect size (Cohen’s f = 0.15). Eight additional subjects were further added for enrolment, given a potential dropout of about 5%, bringing the number of subjects to be enrolled to a total of 156.

Linear random intercept mixed models (LMM) were used to analyse all experimental variables: each of the 10 response variables constituted the dependent variable in independent models, while the measurement in two (T0 and 56 days) or three (T0, 28 days, 56 days) stages, treatments (treated vs. placebo), and their interaction were the fixed factors. Age and gender were considered also as fixed factors to control variations dependent on these effects. Subject identity was included as a random factor to control for individual variability in treatment response. For the secondary endpoints regarding stool consistency and bowel movements, independent analyses were performed for each subgroup using the same model as applied to the variables of the primary endpoint.

Analyses were performed using the lme4 package [20] in R version 4.0.1 [21], unless otherwise specified. Data are reported as means ± standard deviations.

## 3. Results

Figure 1 displays the study flowchart, produced following CONSORT PRO reporting guidelines [21]. Table 1 presents the composition of the examined IBS subgroups (i.e., IBS-C (constipation-predominant), IBS-D (diarrhoea-predominant), IBS-M (mixed)) of the enrolled participants. Recruitment was realised to be representative of the prevalence of each form of IBS in the population. The predominance of IBS-D is therefore reflected in the recruitment of a larger number of subjects for this subgroup (Table 1).

The sample includes 156 subjects for each experimental group (77 women and 79 men). The mean age (±SD) of the subjects was 44 ± 16 years (men: 47 ± 16, women: 41 ± 16), with a range of 18–70 years.

### 3.1. Primary Outcome

The evaluation of the efficacy of the dietary supplement based on a tannin complex in improving the symptoms in patients suffering from IBS was performed using the validated questionnaire administered to the subjects to assess general symptomatology (IBS-SSS). Data were recorded at the beginning of the intervention (T0) and after 56 days. Table 2 reports descriptive statistics (mean, standard deviation, and range) for the comparison between the placebo group and the food supplement group at T0 and 56 days for each of the selected variables.

The findings of the LMM analysis regarding the primary outcomes are displayed in Table 3. The results showed a significant effect for measurement (*p* < 0.001), the experimental group (*p* < 0.001), and their interaction (*p* < 0.001) for all three variables. Conversely, no significant effects emerged for the age and gender of the subject in any of the cases (Table 3).

At T0, subjects of both groups (treated and placebo) fell within the moderate level of symptomatology. Thus, even though at T0, the treated group presented significantly higher IBS-SSS score values than the placebo group (β = 25.5 ± 12.6, t_226_ = 2.349, *p* = 0.043, Figure 2A), the difference was not clinically relevant. Then, the IBS-SSS score values decreased significantly between T0 and 56 days in the treated group (β = 94.1 ± 8.1, t_154_ = 11.64, *p* < 0.001, Figure 2A), whereas it significantly increased in the placebo group (β = 30.1 ± 8.1, t_154_ = 3.723, *p* < 0.001, Figure 2A). Consequently, at 56 days, the IBS-SSS score was significantly lower in the treated group compared to the placebo group (β = 98.2 ± 12.6, t_226_ = 7.867, *p* < 0.001, Figure 2A). This statistically significant difference is relevant from a clinical point of view, as supplementation resulted in a change in the severity of symptoms from moderate to mild.

### 3.2. Secondary Outcomes

The secondary outcomes were aimed at studying both the quality of life of patients and the specific effect on the IBS patient’s intestinal symptomatology.

#### 3.2.1. Patient’s Quality of Life

Table 2 reports descriptive statistics (mean, standard deviation, and range) for the two-variable response, IBS-QoL and GIQLI questionnaires, evaluated at the start and at the end of the intervention.

At T0 the IBS-QoL score values were significantly higher in the placebo group than in the treated group (β = 20.5 ± 0.9, t_306_ = 21.771, *p* < 0.001, Figure 2B). Following the trial, the IBS-QoL score decreased highly significantly in the placebo group (β = 26.3 ± 0.9, t_306_ = 28.107, *p* < 0.001, Figure 2B) and increased significantly in the treated group (β = 5. 5 ± 0.9, t_306_ = 5.841, *p* = < 0.001, Figure 2B). At 56 days, the IBS-QoL score was significantly higher in the treated group than in the placebo group (β = 11.3 ± 0.9, t_306_ = 11.967, *p* < 0.001, Figure 2B). However, the slight increase in the treated group questionnaire score is reflected only as an improving trend at the clinical level.

Similar to the previous situation, the GIQLI score at T0 was significantly higher in the placebo group than in the treated group (β = 10.3 ± 1.0, t_154_ = 10.309, *p* < 0.001, Figure 2C). At the end of the trial, the GIQLI score increased significantly between T0 and 5 in the treated group (β = 43.3 ± 1.0, t_154_ = 43.475, *p* < 0.001, Figure 2C), whereas it decreased significantly in the placebo group (β = 10.3 ± 1.0, t_154_ = 10.309, *p* < 0.001, Figure 2C). At the end of the study, the treated group had significantly higher GIQLI score values than the placebo group (β = 45.2 ± 1.0, t_300_ = 42.28, *p* < 0.001, Figure 2C). At the clinical level, the increase in the questionnaire score indicates a clear improvement in quality of life, which confirms the trend reported above.

#### 3.2.2. Intestinal Symptomatology

Five different specific IBS symptoms were monitored during the whole time of the intervention through the compilation of an intestinal diary. Table 4 reports the descriptive statistics (mean, standard deviation, and range) for the five variables at T0, 28 days, and 56 days. All variables showed the same trend: LMM model statistical analysis found a significant effect for measurement (*p* < 0.001), the experimental group (*p* < 0.001), and their interaction (*p* < 0.001) and no significant effects for age. No significant effect was recorded for gender either, except for the symptom of abdominal pain (*p* = 0.01) (Table 5).

In general, the placebo group showed no major changes during the time of the intervention. Only in the case of the symptom of abdominal pain, while the score did not vary significantly between the baseline and the end of the first month of treatment (β = 0.13 ± 0.08, t458 = 1.652, *p* = 0.10), it decreased significantly in the period ranging from the first month to the end of the treatment (β = 0.21 ± 0.08, t458 = 2.700, *p* = 0.0072) (Figure 3D). However, no significant differences emerged between the baseline and the first month of treatment measurements (β = 0.08 ± 0.08, t458 = 1.047, *p* = 0.29, Figure 3D).

In contrast, in the treated group, no significant differences were observed between IBS symptom measurements at baseline (T0) and the first month of the treatment, for all the variables evaluated. After that, the symptom scores drastically decreased by the end of the intervention, with statistical significance (Figure 3D).

Statistically significant differences are also relevant from a clinical point of view, as the tannin-based supplementation resulted in a decrease in the severity of the assessed symptoms, presenting a one-point decrease on the perceived intensity scale. 

#### 3.2.3. Bowel Movement Frequency and Stool Consistency

Table 6 reports the descriptive statistics (mean, standard deviation, and range) for the frequency of evacuations (bowel movements/week) and stool consistency stratified by IBS-subgroups in the specific subgroups IBS-D, IBS-C, and IBS-M.

The analysis of the number of evacuations for the IBS-D subgroup (Table 7) identified a significant effect for the measurement (*p* < 0.001) and the measurement × treatment interaction (*p* = 0.019). At T0, the treated and placebo groups did not differ (β = 0.96 ± 0.59, t_106_ = 1.618, *p* = 0.11, Figure 4A); following the trial, the number of evacuations decreased significantly for the treated group (β = 1.36 ± 0.28, t_85_ = 4.911, *p* < 0.001), whereas it was similar for the placebo group (β = 0.35 ± 0.32, t_85_ = 1.091, *p* = 0.28). Despite this, at 56 days, the BM did not differ significantly in the two groups (β = 0.04 ± 0.59, t_106_ = 0.078, *p* = 0.94).

For the IBS-C subgroup, the analysis identified a significant effect for measurement (*p* < 0.001), treatment (*p* = 0.0024), and their interaction (*p* = 0.0018) (Table 7). At T0, the number of bowel movements/week did not differ between the placebo and the treated groups (β = 0.87 ± 0.48, t_22_ = 1.816, *p* = 0.083, Figure 4B). Following the trial, the number of evacuations increased significantly in the treated group (β = 2.22 ± 0.28, t_18_ = 7.787, *p* < 0.001). Although there was an increase between T0 and 56 days in the placebo group (β = 0.82 ± 0.26, t_18_ = 3.169, *p* = 0.0053), at the end of the study, the bowel movements/week in the treated group was significantly higher than in the placebo group (β = 2.27 ± 0.47, t_22_= 4.763, *p* < 0.001).

Similarly, the same LMM, applied to the IBS-M subgroup (Table 7), found a significant effect for measurement (*p* = 0.028) and for the measure × treatment interaction (*p* = 0.0013). At T0, the treated group and the placebo group showed no differences (β = 1.27 ± 0.98, t_50_ = 1.306, *p* = 0.20, Figure 4C), but following the trial, the number of evacuations increased significantly for the treated group (β = 1.31 ± 0.36, t_47_ = 3.620, *p* < 0.001), while it remained unchanged for the placebo group (β = 0.27 ± 0.29, t_47_ = 0.922, *p* = 0.36). Finally, at 56 days, the number of bowel movements/week between the treated group and the placebo group did not differ significantly (β = 0.30 ± 0.98, t_50_ = 0.309, *p* = 0.76).

There were no significant effects due to the age and sex of the subject in any of the evaluated cases (Table 7).

Stool consistency was measured at three points, at baseline, 28 days, and at the end of the supplementation, using the Bristol scale score. No significant effects were observed in the subgroups analysed for the experimental treatment, for the measure, for the measure × treatment interaction, or even for the age and sex of the subject (Table 7, Figure 5A–C). The only exception is the IBS-D subgroup, in which a significant effect was found for measurement alone (*p* < 0.001).

## 4. Discussion

The aim of the present study was to test the efficacy of a tannin complex supplement in improving the symptoms of subjects suffering from IBS and their quality of life. The single-centre, randomised, parallel-group, double-blind clinical trial involved the supplementation of 156 subjects enrolled with the treatment or a placebo. After 56 days of treatment, subjects taking the tannin-based supplement showed a significant improvement in gastrointestinal symptoms assessed by the IBS-SSS compared to the placebo group. This improvement also indicates a significantly positive result from a clinical point of view, as the IBS condition was improved from moderate to mild in subjects treated with the food supplement. In contrast, no significant changes were observed in the placebo group.

Since IBS symptomatology is strictly linked to psychological issues, the authors considered it a primary concern to test the effectiveness of the product in the improvement of the subjects’ quality of life. Both self-assessment questionnaires of quality of life, namely IBS-QoL and the GIQLI questionnaire, administered to the subjects highlighted a significant improvement in the treated group following the treatment and in comparison with the placebo group. Although the improvement recorded in the IBS-QoL was moderate, in contrast, the GIQLI score mean value almost corresponded to an excellent quality of life. This discrepancy could be ascribed to the variation in the items that are considered by each questionnaire.

It is important to comprehensively assess the development of gastrointestinal symptoms together with quality-of-life symptoms because IBS is now considered a disorder of gut–brain interaction. Bowel-related symptoms are considered of equal importance to mental symptoms (i.e., depression, anxiety, and somatisation), and both can result in work absenteeism and more medical consultations [22].

Although several types of treatments are available, these show limitations in that they treat a specific single symptom and do not take into account the disorder as a whole. Lately, several national guidelines [23,24] are increasingly promoting more holistic first-line approaches, such as healthy eating habits and lifestyle changes, as they can offer a solution to intestinal and mental issues simultaneously [25]. Similar to nutritional interventions, the tannin complex product tested in the present study clearly shows that it can provide both intestinal and mental relief, representing a possible solution for the management of IBS.

To understand what the underlying mechanisms might be, the previous pilot study by Molino et al. (2024) showed how the same product could positively modulate the composition and activity of the intestinal microbiota of IBS-D patients [26]. Microbiota represents a common denominator bridging the gut and the brain and related symptomatologies [27,28]. In particular, the authors highlighted how tannin supplementation could modulate specific taxa, which have a direct correlation with both intestinal discomfort intestinal symptoms and psychological distress. In addition, the product also showed the ability to modulate microbiota activity by regulating the production of SCFAs. These included a reduction in the production of formate, which has been reported to be responsible for the onset of intestinal bloating and inflammation [29].

Another key aspect is related to altered intestinal permeability in IBS sufferers. An impairment of the intestinal barrier allows the passage of harmful substances to the circulatory stream that would otherwise not pass, leading to chronic inflammation, which in turn causes changes in brain function [30]. The above-mentioned work also reports a significant effect in improving the intestinal permeability of patients with IBS-D [26]. The earlier discussed mechanisms of action may also explain the reduction in symptomatology found through the evaluation of diaries collected from patients. In particular, in the study of Molino et al. (2024), positive modulation of the microbiota showed a correlation with the relief of IBS-specific symptomatology such as abdominal pain and abdominal distension [26].

Other treatments based on nutritional interventions such as dietary plans (i.e., FODMAPs) [6] or probiotics [31,32,33,34] may also partly rely on the rationale of restoring a healthy microbiota; however, the supplement tested in this study also combines specific action on intestinal permeability and modulation of bowel movements.

Alongside the symptomatology common to all forms of IBS, it must be considered that each subtype is marked by specific complaints such as diarrhoea or constipation. This means that nutritional interventions need to be customised in a certain sense, and this is why in more severe cases loperamide is indicated for diarrhoea or polyethylene glycol in the case of constipation [35]. For this reason, the secondary objectives of this clinical trial also focused on investigating the efficacy of the tannin supplement in the specific symptomatology of each IBS subgroup, i.e., stool consistency and frequency of weekly evacuation.

Regarding the stool type score (Bristol scale), there were no significant and/or clinically relevant differences in any of the IBS subgroups for comparison between the treated group and the placebo group at any of the times studied. It should be noted that for all three subtypes assessed, the ratings were within the normal range [36], roughly between 3 and 4.5. Thus, the subjects studied never presented a real issue related to stool consistency.

Conversely, as regards the bowel movements, in the IBS-D subgroup, the number of bowel movements/week, which was initially higher in the treated group than in the placebo group, decreased significantly at the end of the dietary supplement treatment, while it remained unchanged in the placebo group. Thus, the number of weekly evacuations in the treated group and in the placebo group at the end of the respective treatments were similar, although a potentially clinically relevant effect was identified in the reduction of the number of weekly evacuations in IBS-D subjects. The IBS-C subgroup showed a completely opposite behaviour, since its number of weekly evacuations was initially similar in the treated and placebo subjects, increased significantly at the end of treatment with the food supplement, while it remained unchanged in the placebo group, inducing a clinically relevant change.

The astringent action of tannins is well known and therefore several studies and patents report their use for the treatment of IBS-D [37,38,39]. However, if tannins have been extensively studied to counteract diarrhoea, it might seem unusual that they could also be used as a treatment for constipation. There are few studies reporting in the literature this property but limited to a few plant extracts [40,41]. The study by Brown et al. (2016) examining a supplement that contained quebracho extract, together with other phytocomplexes, reported improving effects in patients suffering from constipation [42]. However, none of the studies found in the literature reports an ambivalent action such as that given by the product examined in this study. As reported in the ex vivo study by Mattioli et al. (2023), the components of the supplement used in this study show the remarkable property of modulating intestinal motility. In particular, the two plant extracts derived from chestnut and quebracho showed different effects towards gastrointestinal smooth muscle, thereby regulating its activity [11].

## 5. Strengths and Limitations of the Study

The strengths of the present study lie in the robust study design, which also helped confirm results from a previous pilot study (data not yet published). Furthermore, the study confirmed the tolerability without any adverse effects of the nutritional supplement tested, an aspect of relevance as it is intended for subjects who are particularly susceptible to dietary alterations. While dietary, probiotic and nutraceutical interventions in general for IBS involve relatively long treatment times, the product tested in this study demonstrated efficacy from the first month of supplementation.

As far as the limitations of the study are concerned, as there is no follow-up period, the present study cannot predict the course of the subjects’ symptoms post-treatment. Furthermore, as IBS is a chronic and relapsing disease, it might be interesting to conduct a long-term study involving maintenance supplementation.

## 6. Conclusions

In conclusion, the results of this study highlight the considerable application potential of a chestnut and quebracho tannin complex supplement for the all-round treatment of IBS. Indeed, the synergistic effect of the ingredients showed a marked efficacy for the management of both intestinal symptoms and mental disorders, following administration before main meals (lunch and dinner) for 56 days. Furthermore, as far as we know this is the first randomised clinical trial to highlight the ability of a tannin complex to treat disorders with opposite symptomatology such as diarrhoea and constipation. This suggests the uniqueness of the product studied as it has proven to be safe and effective for the management of all subtypes of IBS, indiscriminately.

## Figures and Tables

**Figure 1 nutrients-17-00552-f001:**
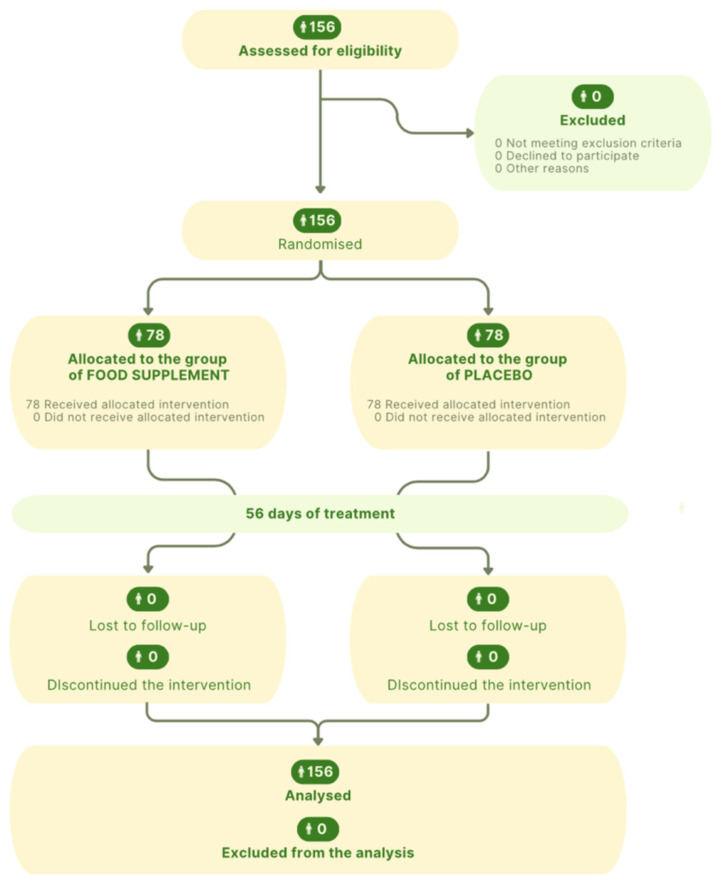
Flow diagram illustrating the enrolment, allocation, follow-up, and analysis phases of participants in the clinical trial, as outlined by the CONSORT (Consolidated Standards of Reporting Trials) guidelines.

**Figure 2 nutrients-17-00552-f002:**
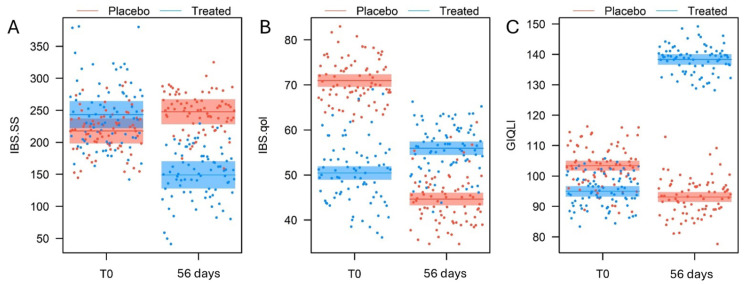
Comparison at T0 and 56 days of the treated and the placebo groups in the primary endpoint (**A**) IBS-SSS and the secondary variable endpoints (**B**) IBS-QoL and (**C**) GIQLI, as predicted by the LMM models (means and 95% confidence intervals).

**Figure 3 nutrients-17-00552-f003:**
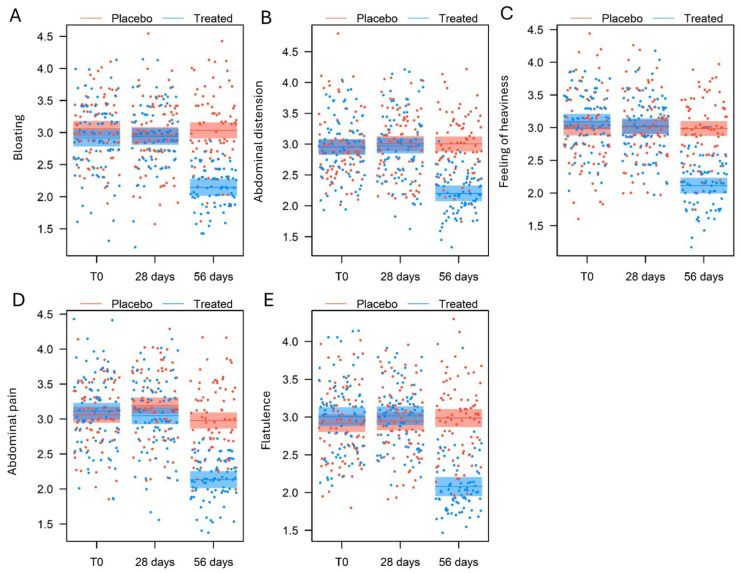
Comparison of the symptoms (**A**) bloating, (**B**) abdominal distension, (**C**) feeling of heaviness, (**D**) abdominal pain, and (**E**) flatulence at T0 (baseline), 28 days, and 56 days of the treated and the placebo groups, as predicted by the LMM models (means and 95% confidence intervals).

**Figure 4 nutrients-17-00552-f004:**
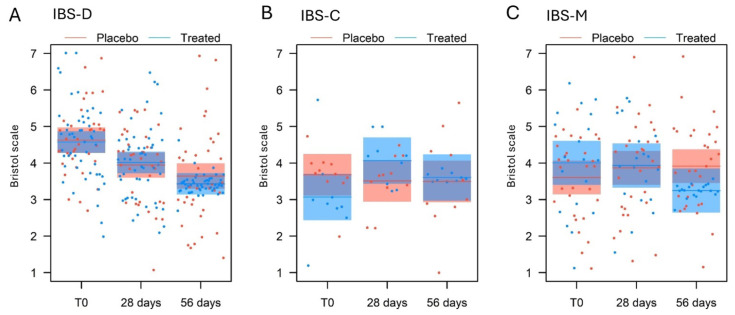
Comparison of the bowel movements per week at T0 (baseline), 28 days, and 56 days of the treated and the placebo groups, for (**A**) IBS-D, (**B**) IBS-C, and (**C**) IBS-M subgroups, as predicted by the LMM models (means and 95% confidence intervals).

**Figure 5 nutrients-17-00552-f005:**
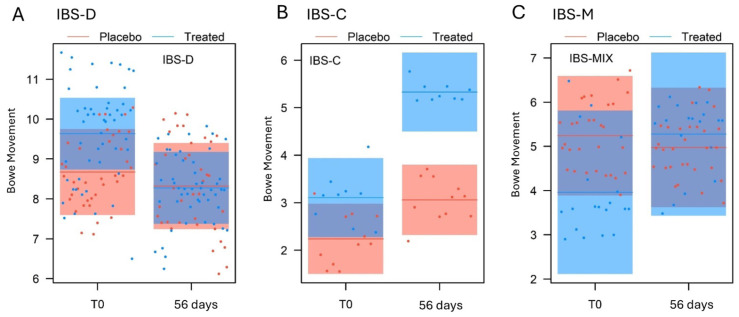
Comparison of the stool consistency at T0 (baseline), 28 days, and 56 days of the treated and the placebo groups evaluated through the Bristol scale, for (**A**) IBS-D, (**B**) IBS-C, and (**C**) IBS-M subgroups, as predicted by the LMM models (means and 95% confidence intervals).

**Table 1 nutrients-17-00552-t001:** Demographic data of the composition of the subgroups examined for the study.

Subgroup	Females (n)	Males (n)	Tot (n)
IBS-C	10	10	20
IBS-D	43	44	87
IBS-M	26	23	49

**Table 2 nutrients-17-00552-t002:** Descriptive statistics (mean, standard deviation, and range of values) for the comparison between the two experimental groups for each of the three variables selected for the primary endpoint, measured at time T0 and at the end of the trial (56 days).

Variable	Placebo		Treatment	
	T0	56 Days	T0	56 Days
IBS-SSS	217.3 ± 80.3	247.4 ± 75.9	242.3 ± 89.8	148.1 ± 60.6
	(50–400)	(80–430)	(60–470)	(40–300)
IBS-QOL	70.6 ± 4.8	44.3 ± 5.8	49.9 ± 7.2	55.3 ± 5.4
	(61–82)	(33–61)	(35–68)	(42–65)
GIQLI	103.7 ± 7.9	93.5 ± 6.9	95.4 ± 6.2	138.7 ± 5.3
	(86–117)	(76–115)	(81–108)	(127–152)

**Table 3 nutrients-17-00552-t003:** LMMs for the primary outcomes.

Variable	F	gdl	*p*
**IBS-SSS**			
Measurement	31.37	1.154	**<0.001**
Treatment	10.73	1.154	**<0.001**
Sex	0.028	1.154	0.87
Age	0.428	1.154	0.51
Measurement × Treatment	118.1	1.154	**<0.001**
**IBS-QOL**			
Measurement	247.9	1.306	**<0.001**
Treatment	47.47	1.306	**<0.001**
Sex	2.332	1.306	0.66
Age	3.365	1.306	0.067
Measurement × Treatment	576.2	1.306	**<0.001**
**GIQLI**			
Measurement	549.9	1.154	**<0.001**
Treatment	523.5	1.154	**<0.001**
Sex	0.701	1.154	0.40
Age	2.774	1.154	0.10
Measurement × Treatment	1446.7	1.154	**<0.001**

**Table 4 nutrients-17-00552-t004:** Descriptive statistics (mean, standard deviation, and range of values) for five response variables—bloating, abdominal distension, feeling of heaviness, abdominal pain, and flatulence—at baseline (T0), after 28 days of treatment, and after 56 days of treatment in the two experimental groups.

Variable	Placebo			Treatment		
	T0	28 Days	56 Days	T0	28 Days	56 Days
**Bloating**	3.1 ± 0.5	3.0 ± 0.6	3.0 ± 0.6	2.9 ± 0.6	2.9 ± 0.5	2.1 ± 0.3
	(2–4)	(2–5)	(2–4)	(1–4)	(1–4)	(1–3)
**Abdominal distension**	2.9 ± 0.5	3 ± 0.5	3.0 ± 0.5	2.9 ± 0.5	2.9 ± 0.6	2.2 ± 0.4
	(2–5)	(2–4)	(2–4)	(2–4)	(2–4)	(1–3)
**Feeling of heaviness**	3.0 ± 0.6	3.0 ± 0.5	3.0 ± 0.5	3.1 ± 0.4	3.0 ± 0.5	2.1 ± 0.4
	(2–4)	(2–4)	(2–4)	(2–4)	(2–4)	(1–3)
**Abdominal pain**	3.0 ± 0.5	3.1 ± 0.5	2.9 ± 0.5	3.0 ± 0.6	3.0 ± 0.5	2.1 ± 0.3
	(2–4)	(2–4)	(2–4)	(2–4)	(1–4)	(1–3)
**Flatulence**	2.9 ± 0.5	2.9 ± 0.5	3 ± 0.5	3.0 ± 0.5	3.0 ± 0.5	2.1 ± 0.4
	(2–4)	(2–4)	(2–4)	(2–4)	(2–4)	(1–4)

**Table 5 nutrients-17-00552-t005:** LMMs for five response variables: bloating, abdominal distension, feeling of heaviness, abdominal pain, and flatulence.

Variable	F	gdl	*p*
**Bloating**			
Measurement	27.20	2.308	**<0.001**
Treatment	48.12	1.152	**<0.001**
Sex	0.052	1.152	0.82
Age	0.211	1.152	0.65
Measurement × Treatment	31.19	2.308	**<0.001**
**Abdominal distension**			
Measurement	27.27	2.308	**<0.001**
Treatment	33.41	1.152	**<0.001**
Sex	1.515	1.152	0.22
Age	1.690	1.152	0.20
Measurement × Treatment	30.13	2.308	**<0.001**
**Feeling of heaviness**			
Measurement	51.69	2.458	**<0.001**
Treatment	31.81	1.458	**<0.001**
Sex	0.428	1.458	0.51
Age	0.825	1.458	0.36
Measurement × Treatment	48.42	2.458	**<0.001**
**Abdominal pain**			
Measurement	64.07	2.458	**<0.001**
Treatment	45.76	1.458	**<0.001**
Sex	6.295	1.458	**0.010**
Age	0.602	1.458	0.44
Measurement × Treatment	35.58	2.458	**<0.001**
**Flatulence**			
Measurement	42.10	2.307	**<0.001**
Treatment	24.88	1.152	**<0.001**
Sex	0.556	1.152	0.46
Age	0.038	1.152	0.85
Measurement × Treatment	53.07	2.307	**<0.001**

**Table 6 nutrients-17-00552-t006:** Descriptive statistics (mean, standard deviation, and range of values) for the comparison between the two experimental groups for each of the bowel movements/week (BM/wk) and stool consistency (Bristol punctuation) selected as the secondary endpoints, measured at baseline (T0), after 28 days, and after 56 days of treatment in the two experimental groups.

	Variable	Placebo			Treatment		
		T0	28 Days	56 Days	T0	28 Days	56 Days
**IBS-D**	**BM/wk**	8.8 ± 2.8		8.5 ± 3.4	9.7 ± 2.9		8.4 ± 1.7
		(4–14)		(3–13)	(1–14)		(4–11)
	**Bristol**	4.7 ± 1.0	4.0 ± 1.0	3.7 ± 1.4	4.6 ± 1.1	4.1 ± 1	3.5 ± 0.3
		(3–7)	(1–6)	(2–7)	(2–7)	(2–6)	(3–5)
**IBS-C**	**BM/wk**	2.2 ± 0.9		3.0 ± 0.6	3.0 ± 1.5		5.2 ± 1.0
		(1–3)		(2–4)	(1–6)		(4–7)
	**Bristol**	3.6 ± 0.6	3.4 ± 0.8	3.4 ± 1.3	2.9 ± 1.2	4.0 ± 0.7	3.5 ± 0.1
		(2–4)	(2–4)	(1–6)	(1–6)	(3–5)	(3–4)
**IBS-M**	**BM/wk**	5.7 ± 3.7		5.4 ± 3.4	4.6 ± 3.3		5.9 ± 1.7
		(1–14)		(2–13)	(1–13)		(4–10)
	**Bristol**	3.7 ± 1.1	3.9 ± 1.3	4.0 ± 1.3	4.1 ± 1.5	4.0 ± 1.3	3.4 ± 0.2
		(1–6)	(1–7)	(1–7)	(1–6)	(2–6)	(3–4)

**Table 7 nutrients-17-00552-t007:** LMMs for the secondary outcomes, divided by the three subgroups IBS-D, IBS-C, and IBS-M.

	Variable	F	gdl	*p*
**IBS-D**	**BW/wk**			
	Measurement	185	16.24	**<0.001**
	Treatment	183	0.679	0.41
	Sex	183	0.184	0.67
	Age	183	0.046	0.83
	Measurement × Treatment	185	5.642	**0.019**
	**Bristol**			
	Measurement	2253	24.82	**<0.001**
	Treatment	1253	0.281	0.60
	Sex	1253	1.009	0.32
	Age	1253	0.994	0.32
	Measurement × Treatment	2253	0.440	0.64
**IBS-C**	**BW/wk**			
	Measurement	118	62.43	**<0.001**
	Treatment	116	12.93	**0.0024**
	Sex	116	0.023	0.88
	Age	116	0.371	0.55
	Measurement × Treatment	118	13.31	**0.0018**
	**Bristol**			
	Measurement	236	1.087	0.35
	Treatment	116	0.008	0.93
	Sex	116	0.614	0.44
	Age	116	0.011	0.92
	Measurement × Treatment	236	2.165	0.13
**IBS-M**	**BW/wk**			
	Measurement	147	5.101	**0.028**
	Treatment	145	0.264	0.61
	Sex	145	1.079	0.30
	Age	145	0.451	0.51
	Measurement × Treatment	147	11.60	**0.0013**
	**Bristol**			
	Measurement	294	0.890	0.41
	Treatment	145	0.095	0.76
	Sex	145	0.666	0.42
	Age	145	0.147	0.70
	Measurement × Treatment	294	2.459	**0.09**

## Data Availability

The original contributions presented in this study are included in the article. Further enquiries can be directed to the corresponding author.

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
