# Peer review of "Improving Irritable Bowel Syndrome (IBS) Symptoms and Quality of Life with Quebracho and Chestnut Tannin-Based Supplementation: A Single-Centre, Randomised, Double-Blind, Placebo-Controlled Clinical Trial"

_nutrients, 2025, doi:10.3390/nu17030552_

Round 1

Reviewer 1 Report

Comments and Suggestions for Authors

Dear authors,
I find it meritorious that you wanted to demonstrate that quebracio + horse nuts can be an important remedy for irritable bowel patients. According to literature data ( Mark Piemmental, UCLA) about 65% of them have SIBO as a cause of IBS. Consequently, the therapeutic efficacy of using tannins is related to combating the overproduction of gas (H2 and CH4) in the small intestine. The results obtained by you in increasing weekly stool frequency is related to blocking intestinal methane synthesis and H2 adsorption, it is clearly known that CH4 induces decreased MMC and cleaning waves in the small bowel. The symptoms that you have indicated in your questionnaire: bloating, abdominal distension are basically the same symptom, and abdominal haviness is not used by the ROMA criteria or current medical practice, in conclusion this analysis of the severity of symptoms should be modified or excluded from the study. I find it hard to believe that none of the 156 patients gave a score of 5 for the severity of their symptoms as I routinely use such questionnaires myself and patients indicate a severity of 5 in at least 10-15% of cases.
The discussion chapter is of poor quality because you did not discuss other studies that have used the same substances in the treatment of patients with IBS such as.
Brown K, Scott-Hoy B, Jennings LW. Response of irritable bowel syndrome with constipation patients administered a combined quebracho/conker tree/M. balsamea Willd extract. World J Gastrointest Pharmacol Ther. 2016 Aug 6;7(3):463-8. doi: 10.4292/wjgpt.v7.i3.463. PMID: 27602249; PMCID: PMC4986399.

Author Response

Comments :  

Dear authors,
I find it meritorious that you wanted to demonstrate that quebracio + horse nuts can be an important remedy for irritable bowel patients. According to literature data ( Mark Piemmental, UCLA) about 65% of them have SIBO as a cause of IBS. Consequently, the therapeutic efficacy of using tannins is related to combating the overproduction of gas (H2 and CH4) in the small intestine. The results obtained by you in increasing weekly stool frequency is related to blocking intestinal methane synthesis and H2 adsorption, it is clearly known that CH4 induces decreased MMC and cleaning waves in the small bowel. The symptoms that you have indicated in your questionnaire: bloating, abdominal distension are basically the same symptom, and abdominal haviness is not used by the ROMA criteria or current medical practice, in conclusion this analysis of the severity of symptoms should be modified or excluded from the study. I find it hard to believe that none of the 156 patients gave a score of 5 for the severity of their symptoms as I routinely use such questionnaires myself and patients indicate a severity of 5 in at least 10-15% of cases.
The discussion chapter is of poor quality because you did not discuss other studies that have used the same substances in the treatment of patients with IBS such as.
Brown K, Scott-Hoy B, Jennings LW. Response of irritable bowel syndrome with constipation patients administered a combined quebracho/conker tree/M. balsamea Willd extract. World J Gastrointest Pharmacol Ther. 2016 Aug 6;7(3):463-8. doi: 10.4292/wjgpt.v7.i3.463. PMID: 27602249; PMCID: PMC4986399.

Response:

Dear Reviewer,

We appreciate your careful analysis of the manuscript and your valuable comments.

Thank you also for your suggestion of the mechanisms of action of tannins in the treatment of IBS symptomatology. Certainly, the theory suggested by you regarding combating the overproduction of gas (H2 and CH4) in the small intestine is largely responsible for the success of this treatment. However, as we know from our previous studies (as also reported in the discussion of the present study), the mechanism of action of the product under investigation is much broader, involving several fronts. Tannins, in fact, play an important function in modulating the composition of the gut microbiota, with an important influence on the gut-brain axis. In addition, another mechanism of action is related to improving intestinal permeability and modulating the immune response (internal data, not yet published).

Regarding the evaluation of symptomatology, we would like to point out that the primary outcome of the study was to evaluate the efficacy of the food supplement in the general improvement of perceived gastrointestinal symptoms. The validated IBSS score questionnaire was used for this purpose. The symptoms you mentioned were evaluated as secondary outcomes, and not always all indicators studied in trials necessarily have to be part of validated questionnaires. Despite this, most of the outcomes assessed are part of indicators widely recognized in the medical field and fall within the group of validated questionnaires.

We agree with the reviewer that sometimes it may seem that bloating and abdominal distension indicate the same disorder. However, more specifically, distension is a form of visible bloating, while bloating is a more general sensation of discomfort that may or may not involve a change in the size of the abdomen. Regarding the feeling of heaviness, we agree with the reviewer that it does not belong to the ROMA criteria, however we felt it was an interesting aspect to take into account, along with the other recognized symptoms and validated questionnaires.

Regarding the deficiency of score 5 in symptom severity, this is due to the fact that the aim of the clinical trial is to evaluate the efficacy of a food supplement that is intended for the improvement of symptoms general population and  for this study the average of the recruited subjects fell into the moderate category of the disorder (IBSS score), so they did not have an extremely pronounced symptomatology.

Finally, we thank the suggestion of including in the discussion chapter the case report of Brown et al. (2016), which has been incorporated into the text. However, it is important to note that in our paper the phytocomplex tested along with quebracho wood extract is chestnut bark extract and not horse chestnut. Horse chestnut and chestnut have a very different composition, the former being characterized mainly by the presence of saponins while the latter by hydrolysable tannins, as indicated in the chapter on materials and methods.

Reviewer 2 Report

Comments and Suggestions for Authors

I congratulate the Authors for this first ever study of its kind. They performed elegant research, resulting in a very good manuscript, with marked practical importance in all types of IBS. Attention was paid to all details.  Introduction nicely introduced the topic and the necessity of this research. Material and methods were described correctly, including questionnaires and the two groups. Exclusion criteria were pertinent. Interventions were well described, with good statistics. Results showed excellent and quick results - efficacy from the first month of administration. Tolerability of the chestnut and quebracho tannin complex was very good. Tables and Figures are of good quality. While many products and interventions have been tried patients with IBS (for each sub-type), including diet and probiotics – but without consistent results, this tannin supplement was proven efficacious in all patients, regardless of constipation or diarrhea. Discussion paragraph is beautifully conceived, with pertinent and personal analysis and comparisons, including possible mechanisms of this tannin supplement. Limitations are included, mainly the short duration of treatment (absence of follow-up phase). Conclusion summarizes perfectly the whole manuscript. The IThenticate report looks fine. I have only minor comments, listed below:

1. Abstract:

a. What was mentioned in Results should be nicely harmonized among Objectives and Methods (The primary outcome of the study focused on the improvements of the IBS symptoms assessed through the validated questionnaire IBS-Severity Scoring System (IBS-SSS). Similarly, secondary outcomes geared towards the improvement of the quality of life and specific IBS symptoms (namely bloating, abdominal distension, feeling of heaviness, abdominal pain and flatulence), were assessed through self-administered questionnaires. Consistency and frequency of depositions (i.e. Bristol scale and bowel movements/week), considered the stratification of patients into subgroups characterised by diarrhoea (IBS-D), constipation (IBS-C), mixed type (IBS-M).).

b. Please insert a sentence also about statistics performed.

c. Results should give concrete data, what was found in numbers, p value etc, as there are plenty of data to present.

d. Conclusions cannot arise from nothing. They have to be based on concrete results (above comment), provided from proper comparison with the placebo group, using the mentioned questionnaires.

2. Keywords: It would be advisable to use other Keywords, not just those belonging to the title. This would increase the likelihood of the paper being found by readers. The importance of Keywords is to improve indexing.

3. Participants and recruiting – I was wondering - Was smoking an exclusion criterion? I found just - “none were taking abuse substances (i.e. caffeine, alcohol, drugs)”.

4. References

a. References [2], [21] and [36] have to written completely.

b. Other two could be added, as they are guidelines by AGA and they concern pharmacological management (by the end of Introduction or in Discussion):

AGA Clinical Practice Guideline on the Pharmacological Management of Irritable Bowel Syndrome With Constipation, by Chang, Lin et al. Gastroenterology 2022.

AGA Clinical Practice Guideline on the Pharmacological Management of Irritable Bowel Syndrome With Diarrhea Lembo, Anthony et al. Gastroenterology 2022.

Author Response

Comments:

I congratulate the Authors for this first ever study of its kind. They performed elegant research, resulting in a very good manuscript, with marked practical importance in all types of IBS. Attention was paid to all details.  Introduction nicely introduced the topic and the necessity of this research. Material and methods were described correctly, including questionnaires and the two groups. Exclusion criteria were pertinent. Interventions were well described, with good statistics. Results showed excellent and quick results - efficacy from the first month of administration. Tolerability of the chestnut and quebracho tannin complex was very good. Tables and Figures are of good quality. While many products and interventions have been tried patients with IBS (for each sub-type), including diet and probiotics – but without consistent results, this tannin supplement was proven efficacious in all patients, regardless of constipation or diarrhea. Discussion paragraph is beautifully conceived, with pertinent and personal analysis and comparisons, including possible mechanisms of this tannin supplement. Limitations are included, mainly the short duration of treatment (absence of follow-up phase). Conclusion summarizes perfectly the whole manuscript. The IThenticate report looks fine. I have only minor comments, listed below:

Response : We would like to thank the editor and the reviewers for their constructive revision of our work. Below we have addressed each of the comments of the reviewers and incorporated their suggestions. For ease of reading, our comments and responses are shown in blue. The changes made to the manuscript are presented in red colour

Comments:

Abstract:

  1. Comment: What was mentioned in Results should be nicely harmonized among Objectives and Methods (The primary outcome of the study focused on the improvements of the IBS symptoms assessed through the validated questionnaire IBS-Severity Scoring System (IBS-SSS). Similarly, secondary outcomes geared towards the improvement of the quality of life and specific IBS symptoms (namely bloating, abdominal distension, feeling of heaviness, abdominal pain and flatulence), were assessed through self-administered questionnaires. Consistency and frequency of depositions (i.e. Bristol scale and bowel movements/week), considered the stratification of patients into subgroups characterised by diarrhoea (IBS-D), constipation (IBS-C), mixed type (IBS-M).).

Response: Thank you for the clarification, we have harmonised the text.

2. Comment: Please insert a sentence also about statistics performed.

Response: A sentence has been added.

3. Comment: Results should give concrete data, what was found in numbers, p value etc, as there are plenty of data to present.

Response: We rewrote the section.

4. Comment: Conclusions cannot arise from nothing. They have to be based on concrete results (above comment), provided from proper comparison with the placebo group, using the mentioned questionnaires.

Response: We rewrote the section.

5. Comment:  Keywords: It would be advisable to use other Keywords, not just those belonging to the title. This would increase the likelihood of the paper being found by readers. The importance of Keywords is to improve indexing.

Response: Thank you for the suggestion. The fact that the keywords also appear in the title is not due to a specific willing, but the words were chosen as the most representative for this work. Anyways we revised this point.

6. Comment: Participants and recruiting – I was wondering - Was smoking an exclusion criterion? I found just - “none were taking abuse substances (i.e. caffeine, alcohol, drugs)”.

Response: Yes, it was also considered an exclusion criterion. It has been added in the text.

7. Comment: References [2], [21] and [36] have to written completely.

Response: Thanks for the note. References [2] and [21] have been revised, adding the address of the web page. Reference [36], on the other hand, was unfortunately taken from a document written in Chinese and no further information can be derived.

8. Comment: Other two could be added, as they are guidelines by AGA and they concern pharmacological management (by the end of Introduction or in Discussion):

AGA Clinical Practice Guideline on the Pharmacological Management of Irritable Bowel Syndrome With Constipation, by Chang, Lin et al. Gastroenterology 2022.

AGA Clinical Practice Guideline on the Pharmacological Management of Irritable Bowel Syndrome With Diarrhea Lembo, Anthony et al. Gastroenterology 2022.

Response: Thanks for the suggestion, the references have been added.

Reviewer 3 Report

Comments and Suggestions for Authors

Journal:  Nutrients (ISSN 2072-6643)

      Manuscript ID: nutrients-3416639

      Health-related quality of life is greatly impacted by irritable bowel syndrome (IBS), a chronic gut-brain axis condition which results in debilitating changes in bowel patterns and recurring abdominal pain. A single-center, double-blind, placebo-controlled clinical study is conducted which was aimed to examine the efficacy of a tannin-complex based supplement in order to improve the symptoms of IBS at the intestinal and brain level. The food supplement was given to the participants for 56 days and with the help of self-administered questionnaires, the IBS symptoms/improvements were assessed. The study further highlights that this food supplement containing tannin was helpful in improving the IBS symptoms and quality of life of the treated subjects as compared to the placebo group. This study further highlights the importance of this supplement (chestnut and quebracho tannin complex) in IBS and demonstrates that the synergistic effect of these ingredients is effective and safe and can be used clinically without any side effects/or and toxicity issues.

      The study is detailed and interesting. The study is helpful in understanding the potential of tannin-based food supplements. However, I have few concerns related to the written draft which are as follows:

Minor concerns:

1-    Line-63, diagnosticated is a right word, please confirm.

2-    Line-64, Rome IV criteria, please explain concisely, what is this criteria, please explain it very concisely in a sentence in continuation with “Rome IV criteria, requiring patient recurrent abdominal pain at least one day per week 64 (on average) in the preceding three months in association with at least two of the follow-65 ing: abdominal pain related to defecation, change in stool frequency, and change in stool 66 form”

3-    Always, introduce space, between a number and the unit e.g. 11%.

4-    Line 104. Full stop is not there before “more in details”.

5-    Line 104-105, More in details, the quebracho colorado (Schinopsis lorentzii Engl.) wood extract is distinguished by the high content in condensed tannins, especially profisetinidins and fisetin, among others.”. Please correct this sentence. It can be “by its high condensed tannins content.”?

6-    Title, “design of the study” is wordy, consider writing it as “Study design”.

7-    Line-126, “in two experimental groups (n = 78, each group, treated group and placebo 126 group)”. It is suggested to change the order of the sentence, as (treated group and placebo group with n=78 per group. As the type of two experimental groups must be highlighted earlier than the number of participants.

8-    Please check the spellings for “colour”.

9-    Line-129, please change the word concealed, choose another word or rewrite the sentence as, was not revealed.

10-  Line 158, please change, consists to consisted of., line-159, allows to allowed.

11- Line-165, “life of the subjects in response to the experimental treatment was carried out through two questionnaires”, can be life of the subjects in response to the experimental treatment which was carried out through two questionnaires.

12- Please consider writing this (0 no discomfort - 5 maximum discomfort) as (0 for no discomfort and for 5 maximum discomfort). 

13- Section 2.6, please bring this whole section after the study design.

14- Line 216 and 217, please correct the sentence, if the subjects were diagnosed with other pathologies, so they were not considered incompatible? Please correct this sentence.

15- Please have space between the symbols and numbers “mild (from 75 to <175), moderate (from 175 to <300) or severe (>300)”. Please correc this throughout the draft.

16-  Line 226, it can be dependent variables. Please check.

17- Line 345, comparison of the symptoms will be the correct sentence. Please correct it.

Major concerns:

1-    Discussion is very long and has been divided into many small paragraphs due to which reader loses focus to clearly understand the outcomes. The discussion paragraphs are not interlinked, for example, Lines 428-431, can be moved/adjusted in the introduction part. I do not see any link of this paragraph here with the rest of the paragraphs. Lines 433-439, please move this paragraph before the strengths and limitations, as this paragraph is more like a conclusion.

2-    Please make a separate title as “Strengths and limitations of the study”, after the discussion section.

3-    It is suggested to make conclusions as a separate heading.

4-    Line-519, effective product, please remove the word product as it sounds a repetition here.

5-    Lines 501-502, the author explains that “Finally, the strengths of the present study lie in the robust study design, which also helped confirm results from a previous pilot study.”, Please cite the pilot study in the reference, and make it clearer, that how the current study confirms results from a previous pilot study, as stated?

6-    It is suggested to please modify/improve the figure 1, “flow diagram” and explain the diagram in a line or two with a short legend.

Author Response

Comments:

Health-related quality of life is greatly impacted by irritable bowel syndrome (IBS), a chronic gut-brain axis condition which results in debilitating changes in bowel patterns and recurring abdominal pain. A single-center, double-blind, placebo-controlled clinical study is conducted which was aimed to examine the efficacy of a tannin-complex based supplement in order to improve the symptoms of IBS at the intestinal and brain level. The food supplement was given to the participants for 56 days and with the help of self-administered questionnaires, the IBS symptoms/improvements were assessed. The study further highlights that this food supplement containing tannin was helpful in improving the IBS symptoms and quality of life of the treated subjects as compared to the placebo group. This study further highlights the importance of this supplement (chestnut and quebracho tannin complex) in IBS and demonstrates that the synergistic effect of these ingredients is effective and safe and can be used clinically without any side effects/or and toxicity issues.

      The study is detailed and interesting. The study is helpful in understanding the potential of tannin-based food supplements. However, I have few concerns related to the written draft which are as follows:

Response: We would like to thank the editor and the reviewers for their constructive revision of our work. Below we have addressed each of the comments of the reviewers and incorporated their suggestions. For ease of reading, our comments and responses are shown in blue. The changes made to the manuscript are presented in red colour

Minor concerns:

  1. Comment: Line-63, diagnosticated is a right word, please confirm.

Response:  The word has been changed for “diagnosed”.

2. Comment: Line-64, Rome IV criteria, please explain concisely, what is this criteria, please explain it very concisely in a sentence in continuation with “Rome IV criteria, requiring patient recurrent abdominal pain at least one day per week 64 (on average) in the preceding three months in association with at least two of the follow-65 ing: abdominal pain related to defecation, change in stool frequency, and change in stool 66 form”

Response: A sentence has been added.

3. Comment: Always, introduce space, between a number and the unit e.g. 11%.

Response: It has been changed.

4. Comment: Line 104. Full stop is not there before “more in details”.

Response: A full stop has been added.

5. Comment: Line 104-105, More in details, the quebracho colorado (Schinopsis lorentzii Engl.) wood extract is distinguished by the high content in condensed tannins, especially profisetinidins and fisetin, among others.”. Please correct this sentence. It can be “by its high condensed tannins content.”?

Response: The sentence has been corrected.

6. Comment: Title, “design of the study” is wordy, consider writing it as “Study design”.

Response: The title has been corrected.

7. Comment: Line-126, “in two experimental groups (n = 78, each group, treated group and placebo 126 group)”. It is suggested to change the order of the sentence, as (treated group and placebo group with n=78 per group. As the type of two experimental groups must be highlighted earlier than the number of participants.

Response: The suggested changes have been made.

8. Comment: Please check the spellings for “colour”.

Response: It is correct as the text is written in British English.

9. Comment: Line-129, please change the word concealed, choose another word or rewrite the sentence as, was not revealed.

Response: The suggested changes have been made.

10. Comment: Line 158, please change, consists to consisted of., line-159, allows to allowed.

Response:  It was decided to keep the present tense in the sentence as the questionnaire was not designed especially for the study. As it is a validated questionnaire, it does not change over time or according to the trial.

11. Comment: Line-165, “life of the subjects in response to the experimental treatment was carried out through two questionnaires”, can be life of the subjects in response to the experimental treatment which was carried out through two questionnaires.

Response: The suggested changes have been made.

12. Comment: Please consider writing this (0 no discomfort - 5 maximum discomfort) as (0 for no discomfort and for 5 maximum discomfort). 

Response: The suggested changes have been made.

13. Comment: Section 2.6, please bring this whole section after the study design.

Response: The section has been moved.

14. Comment: Line 216 and 217, please correct the sentence, if the subjects were diagnosed with other pathologies, so they were not considered incompatible? Please correct this sentence.

Response:  The sentence has been changed.

15. Comment: Please have space between the symbols and numbers “mild (from 75 to <175), moderate (from 175 to <300) or severe (>300)”. Please correc this throughout the draft.

Response: It has been corrected.

16. Comment: Line 226, it can be dependent variables. Please check.

Response: Yes for this reason it has been specified in the next sentence.

17. Comment: Line 345, comparison of the symptoms will be the correct sentence. Please correct it.

Response: The sentence has been corrected.

Major concerns:

1. Comment: Discussion is very long and has been divided into many small paragraphs due to which reader loses focus to clearly understand the outcomes. The discussion paragraphs are not interlinked, for example, Lines 428-431, can be moved/adjusted in the introduction part. I do not see any link of this paragraph here with the rest of the paragraphs. Lines 433-439, please move this paragraph before the strengths and limitations, as this paragraph is more like a conclusion.

Response: Thanks for the advice. We moved the paragraph in section 2.5 of Material and Methods.

2. Comment: Please make a separate title as “Strengths and limitations of the study”, after the discussion section.

Response:  We added a separated section as suggested.

3. Comment: It is suggested to make conclusions as a separate heading.

Response: We added a separated section as suggested.

4. Comment: Line-519, effective product, please remove the word product as it sounds a repetition here.

Response: The word “product” has been removed.

5. Comment: Lines 501-502, the author explains that “Finally, the strengths of the present study lie in the robust study design, which also helped confirm results from a previous pilot study.”, Please cite the pilot study in the reference, and make it clearer, that how the current study confirms results from a previous pilot study, as stated?

Response: The reference of the pilot study was not indicated because it has not been yet published. We added a sentence to specify it.  

6. Comment: It is suggested to please modify/improve the figure 1, “flow diagram” and explain the diagram in a line or two with a short legend.

Response: The figure caption has been improved.

Round 2

Reviewer 1 Report

Comments and Suggestions for Authors

You made changes and responded to all of my suggestions. Thanks. The single thing that will remain unknown is how it is possible to have no patient excluded from initial screening. All 156 agreed to pursue the study design.

Reviewer 3 Report

Comments and Suggestions for Authors

I have no further comments for the authors.